# Uncertainty-Aware Column Generation for Crew Pairing Optimization Using Survival Analysis

## Abstract

The crew pairing problem (CPP) is central to optimal planning and scheduling of operations in the airline industry, where the objective is to assign crews to cover a flight schedule at minimal cost while adhering to various logistical, personnel, and policy constraints. Despite the implementation of optimized schedules, operations are frequently disrupted by unforeseen events. This vulnerability stems from the deterministic nature of the CPP's base formulation, which fails to account for the uncertainties inherent in real-world operations. Existing solutions either aim to safeguard against a specified level of uncertainty or focus on worst-case scenarios.To this end, we propose a reliability-centric CPP formulation amenable to solution by column-generation (CG) `SurvCG`, that leverages survival analysis for dynamic quantification of uncertainty using the operation patterns in historical data. Applied to CPP, `SurvCG` forecasts and incorporates flight connection reliability into the optimization process. Through rigorous experiments on a large-scale first-of-its-kind real-world instance under regular and irregular operating conditions, we demonstrate that `SurvCG` achieves unprecedented improvements (**up to 61%**) over baseline in terms of total propagated delays, establishing `SurvCG` as the first data-driven solution for uncertainty-aware reliable scheduling.

## 1 Introduction

Airline operations planning involves complex decision making on optimal flight scheduling, aircraft assignment, and crew pairing. A pairing is a sequence of flights assigned to a crew under strict rules governed by aviation regulating bodies and union policies. Crew expenses are a major component of airline costs and are highly sensitive to disruptions, since airlines incur additional costs due to any delayed flights, swaps or call-ins (IATA). In this context, the crew pairing problem (CPP) is critical for determining an optimal set of pairings with minimum crew cost under the said constraints.[1]

The CPP is a highly constrained NP-hard combinatorial optimization problem (Aydemir-Karadag et al., 2013; Lu & Gzara, 2015; Deveci & Demirel, 2018), and is typically modelled as a set-partitioning problem, and, the state-of-art solution methods are based on the column generation (CG) method (Zeren & Özkol, 2016; Quesnel et al., 2020). CPP is often solved by minimizing planned costs based on known schedules and flight times assuming no disruption, months prior to the actual day of operations; also used as the benchmark for evaluation, referred to as the *nominal cost* ((Erdoğan et al., 2015; Eltoukhy et al., 2017)). However, unforeseen events, such as crew absenteeism and adverse weather, introduce disruption, making the actual costs considerably higher than the planned ones (Antunes et al., 2019). Such disruptions cause pairings to violate operational constraints, such as union regulations, requiring overtime, crew swaps, or additional crews, often leading to *deadheading* or last-minute crew assignments to maintain coverage, resulting in inflated costs, carbon emissions, and customer dissatisfaction (Huang et al., 2020).

Using historical flight data at the time of planning can help with uncertainty management and crew utilization (Sohoni et al. (2011)). To this end, current approaches introduce uncertainty in CPP by

---

[1]While we use airline operations for this exposition, a wide-range of industries are exposed to uncertain operational environments, including aviation and logistics Ball et al. (2007); Wu (2016), manufacturing Alimian et al. (2019) and healthcare Moosavi & Ebrahimnejad (2018).

Figure 1: `SurvCG` integrates survival analysis with column generation for optimal crew pairing. The reliabilities $r_{ij}$ can be pre-computed or can be queried on-the-fly for the optimization for scalability.

modelling flight delay using predefined intervals or historical data (Lu & Gzara (2015), Antunes et al. (2019)). Nevertheless, they struggle to account for real-time disruptions, since limited information available during planning makes it challenging to accurately predict delays, especially on a new flight route or time. Moreover, while minor prediction errors lead to missed connections, historical averages result in an overly conservative plan, exacerbating operational inefficiencies and increasing costs. Schaefer et al. (2005) used simulation to estimate operational crew costs but didn't incorporate this into optimization, emphasizing the need to consider these costs during planning.

Given this, there is a need to inform the CG-based CPP optimization via data-driven predictions, allowing it to trade-off crew utilization and on-time performance for a selected *reliability*, which virtually leads to planning costs better reflecting operational ones. To this end, we propose `SurvCG` – a survival analysis-based CG algorithm that utilizes survival analysis predicted flight arrival likelihood between two candidate connections (*reliability $r_{ij}$*) in the cost function evaluation; see Fig. 1. Here, the refined cost $\phi(c_p^k)$ captures not only the planned scheduled costs but also historical flight connection reliability – to the best of our knowledge, the first work to incorporate data-driven reliability for this task. Note that, while for this exposition we use this specific cost function, SurvCG's modular structure allows for the use of data-driven reliabilities in other linear and non-linear cost functions. Our evaluations on a real-world on-time performance dataset at different levels of irregular (disrupted) operations reveals that `SurvCG` leads to significant performance gains in total propagated delays, especially in the challenging higher-percentiles of delays, in some cases reducing the delays by $61\%$ – an unprecedented advancement enabled by our data-driven optimization method. This is because, as opposed to historical averages, `SurvCG` can handle the long tail of delay distributions. Our overall contributions may be summarized as follows:

1. **Data-driven reliability-based optimization formulation:** `SurvCG` combines survival analysis with column generation, integrating reliability into the crew pairing optimization, ensuring schedules are rigorously penalized for low reliability, thus optimizing for both efficiency and robustness. To the best of our knowledge, this is the first approach to explicitly quantify real-world uncertainties using time-to-event models.

2. **Introduce P-Index to measure the predictive ability of time-to-event models.** Conventional survival metrics (e.g., C-index, Brier score) only consider event ordering, inadequate when exact event timing is crucial. We propose the P-index to assess model precision.

3. **Instance generation and rigorous analysis at different levels of disruptions:** Using real-world flight data, we generate instances and run extensive simulations to establish the superior properties of `SurvCG` under various operational conditions. Our public dataset-based instance offers the first benchmark for this task to drive advancements in the area. https://anonymous.4open.science/r/SurvCG-Instance-67C6/

## 1.1 RELATED WORKS

Traditional crew pairing models are deterministic and fail to account for disruptions such as weather, delays, or maintenance issues. To this end, stochastic programming has been applied to introduce

randomness and develop more disruption-resilient solutions. For instance, Yen & Birge (2006) aimed to minimize expected total costs through stochastic programming, though scalability remains an issue. Ionescu & Kliewer (2011) and Dück et al. (2012) extended this work by incorporating crew swaps and combining crew scheduling with aircraft assignment for operational resilience. Schaefer et al. (2005) estimate crew pairing costs via simulation, without reflecting this in the optimization. They propose heuristic-based improvements over nominal solution using penalties on undesirable features.

Robust optimization approaches, which account for uncertainty by modeling worst-case scenarios, have also been explored. These models introduce additional constraints and variables, increasing computational complexity. Antunes et al. (2019) developed a robust crew pairing model that accounts for delay propagation and operational disruptions, while Lu & Gzara (2015) proposed a robust optimization approach using Lagrangian relaxation to handle crew costs under worst-case conditions. However, these methods often rely on historical delays or predefined uncertainty sets. This is limiting because they averages are not sufficient to capture the long-tail distribution of delays, and therefore these methods' ability to handle irregular disruptions in the real-world.

Time-to-event modeling is used in a number of domains while survival analysis is a popular choice in clinical studies to analyze disease progression Collett (2015); In & Lee (2018); George et al. (2014), in engineering it is referred to as reliability engineering; we use survival analysis w.l.o.g. since recent developments use this terminology. Survival analysis has been successfully applied in traffic incident modeling Nam & Mannering (2000); Hojati et al. (2014); Li et al. (2020) and predictive maintenance Vianna & Yoneyama (2017); Verhagen & De Boer (2018). A standard approach in survival analysis is to use the Cox proportional hazard (CoxPH) model Cox (1972)], which is a semi-parametric model that assumes that 1) the logarithm of risk (*hazard*) of an event has a linear dependence on their covariates Breslow (1975), and 2) hazard of two data samples remains constant over time, known as the *proportional hazard assumption*. The linearity and the proportional hazard assumptions are limiting in real-world applications, and works such as Liestbl et al. (1994); Faraggi & Simon (1995); Ishwaran et al. (2008a), and more recently neural network-based models Katzman et al. (2018); Lee et al. (2018); Zhong et al. (2021) have become popular.

In the context of aviation, survival analysis has been used to analyze individual flight delays, such as factors influencing delay recovery in an airline Wong & Tsai (2012), and assess delays in South Korea's air transportation Kim & Bae (2021). To the best of our knowledge, time-to-event models have not been used to predict connection reliability or in any optimization settings.

## 2  SURVIVAL ANALYSIS FOR FLIGHT CONNECTION RELIABILITY

This section develops time-to-event (TTE) terminology for survival analysis in the context of flight events, such as arrivals and departures, with examples. We then introduce flight connection reliability forecasting based on a flight's likelihood of arriving within the necessary time window.

### 2.1  METHODOLOGY FOR RELIABILITY PREDICTION

The success of a connection depends on whether the flight $i$ lands within a feasible window before flight $j$'s departure, to allow crew to transition to the next flight. Consequently, delays in flight $i$ can disrupt the entire sequence to be completed by a crew, causing reassignments or missed connections. Hence, we will define the reliability based on flight $i$'s timely arrival within the connection window. To this end, we use survival analysis to determine the probability that a flight $i$ can connect to a subsequent flight $j$ departing at scheduled departure time, $\text{SDT}_j$. The time-to-event (TTE) for flight $i$ is defined as $\text{TTE}_i = \text{AAT}_i - \text{SDT}_i$, where $\text{AAT}_i$ is the actual arrival time, demonstrated as follows.

> **Example 2.1**
>
> Consider a flight from New York (JFK) to Los Angeles (LAX) with a ($\text{SDT}_i$) of 08:00 AM and an ($\text{AAT}_i$) of 11:30 AM, both in Central Time (CT). TTE for this flight would be $\text{TTE}_i = 11:30\,\text{AM} - 08:00\,\text{AM} = 3.5\,\text{hours}$. If the subsequent flight $j$ from LAX to San Francisco (SFO) is scheduled to depart at 12:00 PM ($\text{SDT}_j$), we need to determine if the connection between these flights is feasible given the TTE.

We represent each flight, $i$ by a tuple $(\mathbf{z}_i, y_i, D_i)$, where $\mathbf{z}_i \in \mathbb{R}^d$ has features like origin, destination, $\text{SAT}_i$, aircraft type/model etc., $y_i$ is the TTE, and $D_i$ is the event indicator. The event $D_i$ is 1 if the flight lands by time $t$, otherwise 0. The survival function for a flight with features $\mathbf{z}$ at time $t$ is:

$$
\begin{aligned}
S(t \mid \mathbf{z}) &= \mathbb{P}(\text{flight landing beyond time } t \mid \text{flight's features } \mathbf{z}) &&= \mathbb{P}(T > t \mid \mathbf{Z} = \mathbf{z}) \\
&= 1 - \mathbb{P}(\text{flight landing within time } t \mid \text{flight's features } \mathbf{z}) &&= 1 - \mathbb{P}(T \leq t \mid \mathbf{Z} = \mathbf{z}) \quad (1)
\end{aligned}
$$

Here, $\mathbf{Z}$ and $T$ are random variables corresponding to features $\mathbf{z}$, and the associated TTE.

To predict the reliability $r_{ij}$ of two flights $i$ and $j$ in a sequence, we aim to estimate the probability that flight $i$ will arrive in time for the subsequent flight $j$ to depart. Specifically, we need the probability $r_{ij}$ that the flight $i$ to land in time $t_q = \text{SDT}_j - \text{SDT}_i - \delta_{min}$, where $\delta_{min}$ is minimum sit time between flights. Using (1), we estimate $r_{ij}$ by querying the estimated survival function $\widehat{S}(t \mid \mathbf{z_i})$ as

$$
r_{ij} = \mathbb{P}(T \leq t_q \mid \mathbf{Z} = \mathbf{z}_i) = 1 - \widehat{S}(t_q \mid \mathbf{z}_i), \text{where } t_q = \text{SDT}_j - \text{SDT}_i - \delta_{min}. \quad (2)
$$

The survival function $\widehat{S}(t)$ can be estimated using a non-parametric Kaplan & Meier (1958) estimator from empirical data as follows, where $t_1, t_2, \ldots, t_L$ are unique times of flight landing, $d_i$ denotes the flights that landed at time $t_i$, $n_i$ be the flights that could possibly land at time $t_i$ and $\mathbb{1}\{\cdot\}$ is the indicator function. However, this cannot be used at time points without event observations.

$$
\widehat{S}(t) = \prod_{i=1}^{L} \left( 1 - \frac{d_i}{n_i} \right)^{\mathbb{1}\{t_i \leq t\}}, \text{where } d_i = \sum_{j=1}^{n} \mathbb{1}\{y_j = t_i\} D_j, n_i = \sum_{j=1}^{n} \mathbb{1}\{y_j \geq t_i\} \quad (3)
$$

Therefore, semi-parametric methods, such as CoxPH Cox (1972) and DeepSurv Katzman et al. (2018) which can capture non-linearity when modeling covariates, are a popular choice to tackle such cases since since these can provide continuous survival estimates that extend beyond the observed event times. CoxTime Kvamme et al. (2019) further extends the CoxPH model by allowing the risk score to vary with time. The hazard function is defined below, where $f(\mathbf{z}, t; \theta)$ is a time-dependent neural net.

$$
h(t \mid \mathbf{z}) = h_0(t) \exp\left( f(\mathbf{z}, t; \theta) \right). \quad (4)
$$

This model relaxes the proportional hazards assumption by allowing the effect of covariates on hazard to vary over time. We found CoxTime's performance to be competitive and hence use it for this exposition. In general, any time-to-event survival model which preserves the probability interpretation can be used with `SurvCG`; See (Moore, 2016; Freedman, 2008) for a primer on survival analysis.

## 2.2 P-INDEX: EVALUATING THE PREDICTIVE PERFORMANCE OF SURVIVAL MODELS

The most commonly used evaluation metric for survival models is concordance index or C-index Harrell et al. (1982). It quantifies the rank correlation between the actual time-to-event (TTE) and the model's predictions. However, C-index has been found to be less effective for evaluating models that violate the proportional hazards' assumption Antolini et al. (2005). Hence, we first use the time-dependent C-index, $C^{td}$ given by Antolini et al. (2005), as used in Kvamme et al. (2019). For an estimate $\hat{S}(t \mid \mathbf{z})$, $C^{td}$ estimates the probability that the predicted TTE $T_i$ for flight $i$ is less than the TTE $T_j$ for flight $j$, given that $T_i$ is less than or equal to $T_j$ as

$$
C^{td} = P(\hat{S}(T_i \mid \mathbf{z}_i) < \hat{S}(T_j \mid \mathbf{z}_j) \mid T_i \leq T_j, D_i = 1). \quad (5)
$$

However, since our interest extends beyond the discriminative ability of the model; we focus on the accuracy with which probabilities derived from the predicted survival curve are mapped to specific times. As illustrated by (2), the reliability of a connection $r_{ij}$ is obtained by querying a time $t_q$ against the predicted survival function $\widehat{S}(t \mid \mathbf{z})$. This mapping is crucial, as errors in the probability estimation for specific times impact the decision to choose a connection over the other. Therefore, to evaluate the accuracy of this mapping, we introduce $P$-index, defined as:

$$
P\text{-index} = \frac{\sum_{n=1}^{N} \mathbb{1}(|\hat{t}_q^n - t_q^{*,n}| \leq \varepsilon)}{n}. \quad (6)
$$

Here, with a discretization $N$ of a predicted survival function for the $i$-th flight (with certain origin, destination, and $\text{TTE}_i$), for each $n \in N$, the indicator is 1 if the predicted query time $\hat{t}_q^n$ is within a

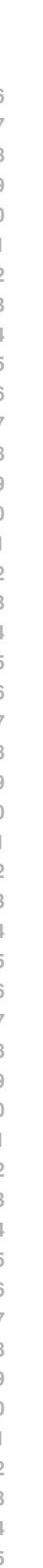
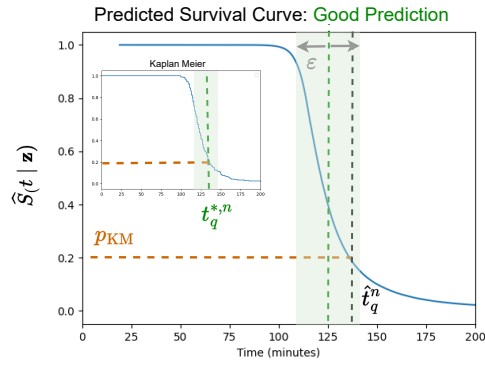
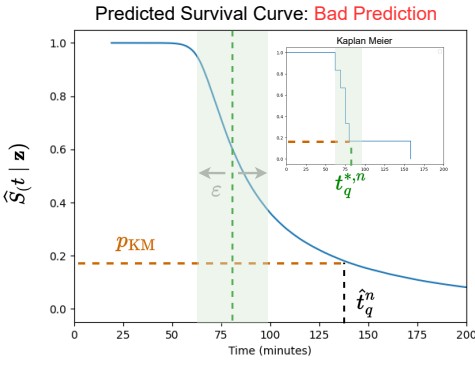

(a) Within: $\hat{t}_q^n$ for flight 1 is within $\varepsilon$

(b) Not within: $\hat{t}_q^n$ for flight 2 is outside $\varepsilon$.

Figure 2: P-index: Ratio of predictions within $\varepsilon$ to total predictions across query times $t_q$.

predefined error margin $\varepsilon$ of the actual query time $t_q^{*,n}$, and 0 otherwise. Here, $t_q^{*,n}$ is the estimate of the event time for flight $i$ based on the non-parametric Kaplan-Meier estimator $\hat{S}_{KM}(t_q^*|\mathbf{z}_i)$. The $\hat{t}_q^n$ is determined by the time at which the estimated survival probability by the model $\hat{S}(t|\mathbf{z}_i)$ matches the Kaplan-Meier survival probability $p_{\text{KM}} = \hat{S}_{KM}(t_q^{*,n}|\mathbf{z}_i)$. Figure 2 illustrates cases where $\hat{t}_q^n$ is within and not within $\varepsilon$. Additionally, to quantify the deviations we compute the Mean Absolute Error (MAE) for predictions within, above, and below the actual query times:

$$\text{MAE}_l = \frac{\sum_{i=1}^n |\hat{t}_q^n - t_q^{*,n}| \cdot \mathbb{1}_\theta}{\text{card}(\theta)}, \quad \theta \in \{\text{Within}(\equiv), \text{Above}(>), \text{Below}(<)\} \tag{7}$$

where $\mathbb{1}_\theta$ is the indicator function for each condition $\theta$, and $\text{card}(\theta)$ is the number of instances satisfying the condition $\theta$. Above represents $\hat{t}_q^n > t_q^{*,n}$, and Below represents $\hat{t}_q^n < t_q^{*,n}$.

# 3 SURVIVAL-BASED COLUMN GENERATION (SURVCG)

We model the crew pairing problem for a set of crews $K$ on a flight network $G = (N, A)$; where $N$ includes origin and destination nodes $O$ and $D$ representing the crew base at the start and end of the schedule, respectively, and flight nodes $N \setminus \{O, D\}$ corresponding to the set of flights $\mathcal{F}$ in the schedule. A flight node is defined by the flight's origin and destination airports, and respective departure and arrival times. The set $A$ comprises three types of arcs: arc $(O, i)$ if crew $k$ can start its schedule with flight $i$, arc $(i, D)$ if crew $k$ can end its schedule with flight $i$, and arc $(i, j)$ for sequential flights $i$ and $j$ where the destination airport of $i$ is the same as the origin airport of $j$ and the minimum sit time is met. The latter is the minimum time needed for the crew to transition between two consecutive flights in their pairing. The set $P^k$ represents all possible pairings for crew $k \in K$, where pairing $p \in P^k$ has an associated cost $c_p^k$. The binary parameter $a_{ip}$ is 1 if flight $i \in \mathcal{F}$ is covered by pairing, $p \in P^k$ and 0 otherwise. The binary decision variable $x_p^k$ equals 1 if pairing $p \in P^k$ is selected for crew, $k \in K$ and 0 otherwise. The set-covering formulation of the reliable crew pairing problem RCPP is given by:

$$[\text{RCPP}]: \text{minimize} \quad \sum_{k \in K} \sum_{p \in P^k} \phi(c_p^k) \tag{8}$$

$$\text{subject to} \quad \sum_{k \in K} \sum_{p \in P^k} a_{ip} x_p^k \geq 1 \quad \forall i \in \mathcal{F} \tag{9}$$

$$\sum_{p \in P^k} x_p^k = 1 \quad \forall k \in K \tag{10}$$

$$x_p^k \in \{0, 1\} \quad \forall k \in K, \forall p \in P^k \tag{11}$$

The objective function (8) minimizes the total cost of selecting $|K|$ crew pairings using the reliability-integrated cost function $\phi(.)$. Constraints (9) ensure that each flight $i \in \mathcal{F}$ is covered by at least one

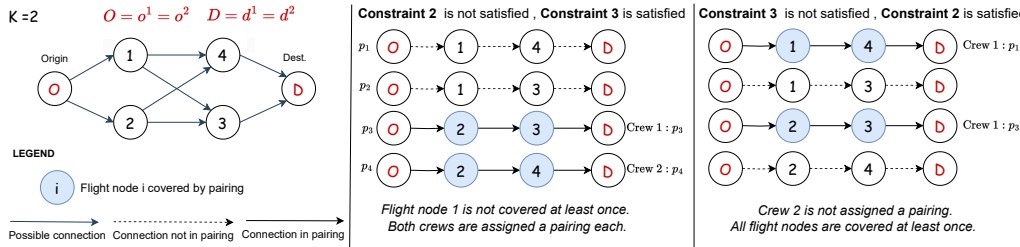

Figure 3: Network and Constraints: The first panel shows the flight network. The middle one violates Constraint 2 (node 1 uncovered), while the last violates Constraint 3 (Crew 2 without pairing).

pairing. Constraint (10) makes sure that each crew $k \in K$ is assigned exactly one pairing from their respective set $P^k$. Figure 3 illustrates constraint effects on a 4-flight, 2-crew network.

The reliable cost function $\phi$ reflects not only the cost of the pairing but also the reliability of the flight connections within. The cost of a pairing $c_p^k$ for pairing $p$ and crew $k$, and its reliability-integrated cost, $\phi(c_p^k)$ are defined on the arcs $A$ of the flight network as follows, with the cost of including arc $(i, j)$ being $c_{ij}$ in pairing $p \in P^k$; the explicit expressions of these costs are detailed next.

$$c_p^k = \sum_{(i,j)\in p} c_{ij}, \quad \forall p \in P^k; \quad \phi(c_p^k) = \sum_{(i,j)\in p} \phi_{ij}(c_{ij}), \quad \forall p \in P^k. \tag{12}$$

### 3.1 NOMINAL VS RELIABLE COST FUNCTIONS

The cost of arc $(i, j)$, $c_{ij}$, corresponds to the cost of the crew covering flight $j$ after flight $i$ in their pairing, and is calculated apriori in function of scheduled departure and arrival times. As such, it is referred to as the nominal cost to distinguish from the actual crew cost, which is based on actual departure and arrival times. $c_{ij}$ is calculated based on the elapsed time of flight $i$ and the connection time between the two consecutive flights $i$ and $j$, expressed as:

$$c_{ij} = (c_i^e + \alpha \cdot c_{ij}^c), \text{where } c_i^e = \text{SET}_i, c_{ij}^c = \text{SDT}_j - \text{SAT}_i \tag{13}$$

The scaling factor $\alpha$ penalises longer connection times, making pairings with shorter layovers between flights more attractive from a cost perspective. Specifically, $\alpha$ reflects the operational priorities, such as reducing crew downtime or enforcing extended layovers. The nominal cost function 13 is not equivalent but mimics the pay-and-credit model, which itself does not accurately reflect the complexity of crew pay in practice. Our function is motivated by discussions with an industry partner.

The nominal cost $c_{ij}$ assumes perfect operation of the airlines, which is rarely the case. Delays in one flight may cause a cascading effect of delays and disruptions in subsequent flights in the pairing, leading to actual costs significantly different from the nominal. The reliable cost function $\phi_{ij}(c_{ij})$ that we propose makes use of the reliability score of a connection to augment the nominal cost and account for delays and disruptions under a push-back recovery policy that is commonly used in the literature (Schaefer et al. (2005); Antunes et al. (2019); Lu & Gzara (2015)). It is expressed as:

$$rc_{ij} = \phi_{ij}(c_{ij}) = c_{ij}(\lambda_1 e^{-\lambda_2 r_{ij}} + 1) - c_{ij}(\lambda_1 e^{-\lambda_2}) \tag{14}$$

where $\lambda_1$ is a parameter that adjusts how significantly the reliability score impacts the reliable cost. A higher value of $\lambda_1$ increases the sensitivity of the cost adjustment to changes in $r_{ij}$. The parameter $\lambda_2$ controls the rate of exponential decay, i.e, the rate of increase of the reliable cost as $r_{ij}$ decreases. A larger $\lambda_2$ results in a steeper decay curve, which more aggressively penalizes lower $r_{ij}$ values. The term $c_{ij}(\lambda_1 e^{-\lambda_2})$ is the vertical adjustment, it shifts the cost function such that the reliable cost equals the nominal cost under ideal conditions ($r_{ij} = 1$). The effects of varying $\lambda_1$ and $\lambda_2$ on the reliable cost function are illustrated in Figure 4.

The reliable cost function $\phi_{ij}(c_{ij})$ captures the trade-offs between reliability and cost efficiency and is nonlinear in $c_{ij}$. Furthermore, similar incorporation of reliability is possible into a pay-and-credit costing model. Consequently, $\phi(c_p^k)$ may be calculated for a given pairing $p \in P^k$ and the objective function of [RCPP] remains linear in the decision variable $x_p^k$. Hence, the linear programming

Table 1: Comparison of survival models. $\eta$ is the learning rate, and $B$ is the batch size. $C^{td}$ is the time-dependent C-Index. CoxTime shows the best performance across all metrics. MAE in minutes.

| Model | $\eta$ | $B$ | $C^{td}$ | $P$-index | MAE$_{\equiv}$ | MAE$_>$ | MAE$_<$ |
|---|---|---|---|---|---|---|---|
| CoxPH | 0.001 | 16 | 0.787 | 0.494 | 8.057 | 38.594 | 10.261 |
| DeepSurv | 0.0001 | 64 | 0.795 | 0.532 | 8.440 | 48.374 | 8.090 |
| CoxTime | 0.0001 | 32 | **0.807** | **0.911** | **4.839** | **21.227** | **4.730** |

relaxation of [RCPP] may be solved by column generation, where the subproblem is a shortest path problem with modified reliable costs on the arcs. Once the relaxation is solved, one has to apply branch-and-price in order to obtain the optimal solution. It is known that solving [RCPP] restricted on the set of generated pairings by CG is usually optimal or very close to optimal. In our experiments, we observed an optimality gap around 0.05%. We note that both [RCPP] and its nominal version CPP defined on the nominal costs are solved using CG. All the instances with varying cost function and parameter settings were solved using CPLEX v22.1.1 using docplex python API on a resouce with 4 CPU cores, 25 GB RAM, and all computational times were less than 30mins.

## 4 EXPERIMENTAL RESULTS

This section validates `SurvCG` by outlining the dataset, hyperparameter tuning, and model performance. It details instance generation, solution comparisons, and simulations, providing quantitative evidence of the approach's robustness and effectiveness.

### 4.1 SURVIVAL MODEL TESTING AND RELIABILITY PREDICTION

**Dataset:** We train the survival analysis model using the Bureau of Transportation Statistics (2024) (BTS) On-Time Performance dataset for flight operations of Endeavor Air in 2019. There are 97294 total flights, of which 80% are used for training and 20% are used for testing. Specifically, for each flight $i$ in the data, the feature set $\mathbf{z}_i$ consists of spatiotemporal attributes and aircraft information including the day of week, aircraft age/model, origin, destination, and scheduled departure/arrival.

**Model Implementation:** We implement DeepSurv (Katzman et al., 2018), CoxPH (Cox, 1972), and CoxTime (Kvamme et al., 2019). DeepSurv and CoxPH assume proportionality; CoxTime doesn't.

**Hyperparameter Tuning + Performance Comparison:** We perform hyperparameter tuning for each model to find an optimal combination of learning rate $lr$, and batch size $b$. For evaluation, we primarily consider the P-index(6) for evaluating model performance and also consider the $C^{td}$ (5). The optimal hyperparameter configurations and evaluation results are shown in Table 1.

**Reliability Prediction:** We use CoxTime (Kvamme et al. (2019)) for reliability prediction due to its superior performance across all quantitative metrics. Notably, CoxTime achieved an $P$-index of 0.911, which indicates its robust capability to accurately predict survival functions. Further, this precision is crucial for predicting if delayed flights will meet minimum crew connection times $\delta_{min}$.

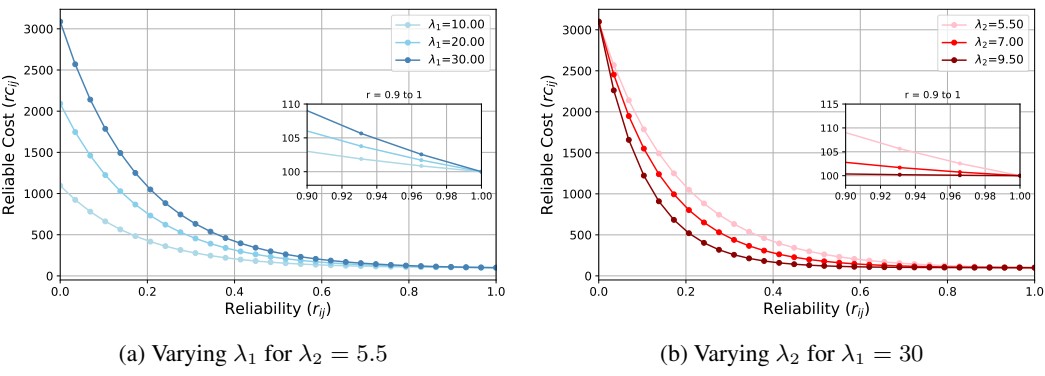

(a) Varying $\lambda_1$ for $\lambda_2 = 5.5$     (b) Varying $\lambda_2$ for $\lambda_1 = 30$

Figure 4: Effects of varying $\lambda_1$ and $\lambda_2$ on the Reliability Integrated Cost Function

## 4.2 COMPARISON OF SOLUTIONS

**Setup:** Given an instance (see Appendix C for detailed instance generation), we run the CG algorithm for the nominal cost function with $\alpha = 2$, as defined in equation 13, meaning that the sit time is penalized twice compared to the elapsed time. For the reliable solutions, we use two configurations with $\lambda_1 = 10, \lambda_2 = 3$ and $\lambda_1 = 20, \lambda_2 = 4$, as defined in equation 14.

**Cost comparison:** We compute the cost of the solutions based on the nominal arc cost (13) with $\alpha = 1$ to ensure nominal and reliable solutions are comparable. The constraint (9) allows for a flight node to be covered more than once, resulting in deadheading, where crew members are transported as passengers, incurring additional costs. Reliable solutions show an increase in deadhead flying costs by up to 5.93% compared to the nominal solution. However, these reliable solutions exhibit significant reductions in deadhead connection costs, with decreases up to 13.58%. This results in marginally lower total costs by up to 0.003%, as seen in Table 2. In comparison, Antolini et al. (2005) report a 1-3% increase in planned costs for their robust solutions.

The number of deadheads in the nominal $\alpha = 2$ solution is 28, based on the flight frequency count (230 nodes covered once, 2 nodes twice, 9 nodes thrice, and 2 nodes four times). For the reliable solution $\alpha = 2, \lambda_1 = 10, \lambda_2 = 3$, the number of deadheads is 24 (232 nodes covered once, 7 nodes thrice, 2 nodes four times, and 2 nodes five times). The configuration $\alpha = 2, \lambda_1 = 20, \lambda_2 = 4$ also results in 24 deadheads. The deadheading is lower for reliable solutions, as reflected by deadhead costs. Additionally, severity of deadheading into a node is also much lower for reliable solutions.

## 4.3 SIMULATION

**Setup:** Given a crew pairing solution, either nominal or reliable ($\lambda_1 = 20, \lambda_2 = 4$), where a pairing $p \in P_{opt}$ covers flights $\mathcal{F}_p$, we simulate by obtaining an actual elapsed time for $i \in \mathcal{F}_p$. The simulation follows Antunes et al. (2019), where the actual elapsed time $\text{AET}_i$ is given by:

$$\text{AET}_i = \text{SET}_i + \epsilon_i, \quad \epsilon_i \sim \text{Kernel Density Estimation}_i^{\text{arrival delay}} \text{ on matched flights} \quad (15)$$

For details on how matched flights are identified, refer to Appendix D.

**Design of Experiments:** We simulate across multiple scenarios by varying two controls: the percentage of irregular operations and the severity of these delays/irregular operations. These variables determine from where the $\epsilon_i$ will be sampled for the matched flights. Each flight is simulated across 100 runs. The percentage of irregular operations (% IR) indicates the proportion of runs (realizations of the pairings) that experience irregularities, while the level of delay specifies the severity of these irregularities. Delay values for irregular operations (IR) are sampled from a specified percentile of the delay distribution using Kernel Density Estimation (KDE). These parameters govern how the flight delays $\epsilon_i$ are sampled for each realization. Scenarios are denoted using the format $m\text{R},$ $n\text{IR-}L$, where $m\%$ of the runs are regular, $n\%$ are irregular, and $L$ represents the percentile beyond which delays are considered. For a full description of the scenarios, refer to Appendix F.

**Metrics:** We evaluate simulation outcomes using Total Propagated Delay (TPGD), which quantifies delays carried from one flight segment to the next, capturing the cascading effects of delays.

$$\text{TPGD} = \sum_i \text{pgd}_i, \quad \text{pgd}_i = \begin{cases} \Delta - (\text{SDT}_{i+1} - \text{AAT}_i), & \text{if } \text{SDT}_{i+1} - \text{AAT}_i < \Delta \\ 0, & \text{otherwise} \end{cases}$$

Table 2: Comparison of Deadheading and Total Costs for Nominal (N)and Reliable (R) solutions ($\alpha = 2$). Changes in costs (in parentheses) are percentages compared to the nominal solution. Headers: DFC - Deadhead Flying Cost, DCC - Deadhead Connection Cost, TFC - Total Flying Cost, TCC - Total Connection Cost, TC - Total Cost. R1- $(\lambda_1, \lambda_2) = (10, 3)$, R2- $(\lambda_1, \lambda_2) = (20, 4)$.

| Solution | Deadheading Cost | | Total Cost | | |
| | DFC | DCC | TFC | TCC | TC |
| --- | --- | --- | --- | --- | --- |
| Nominal | 2948.0 | 17436.0 | 29233.0 | 214933.0 | 244166.0 |
| Reliable 1 | 3123.0 (5.93) | 16155.0 (-7.35) | 29408.0 (0.60) | 214750.0 (-0.09) | 244158.0 (-0.003) |
| Reliable 2 | 3123.0 (5.93) | 15015.0 (-13.58) | 29408.0 (0.60) | 214750.0 (-0.09) | 244158.0 (-0.003) |

Table 3: Total Propagated Delays for 75R,25IR Scenarios with $L = (70, 80, 90)$; pth represents the pth percentile of the TPGD (Total Propagated Delays); N: Nominal, $R$: Reliable, with percentage change in Reliable relative to Nominal. Improvements where R < N are highlighted.

| pth | 75R,25IR-70 | | 75R,25IR-80 | | 75R,25IR-90 | |
|-----|------|------|------|------|------|------|
| | N | R | N | R | N | R |
| 90 | 358.80 | 394.40 (9.94% ↑) | 476.00 | 515.70 (8.37% ↑) | 1080.90 | 831.10 (-23.11% ↓) |
| 91 | 382.65 | 409.07 (6.90% ↑) | 583.91 | 680.31 (16.53% ↑) | 1119.24 | 922.99 (-17.54% ↓) |
| 92 | 461.76 | 430.40 (-6.79% ↓) | 858.80 | 734.00 (-14.53% ↓) | 1155.52 | 939.24 (-18.75% ↓) |
| 93 | 484.52 | 437.03 (-9.80% ↓) | 891.98 | 745.62 (-16.39% ↓) | 1218.96 | 1011.49 (-17.02% ↓) |
| 94 | 535.16 | 480.20 (-10.27% ↓) | 905.96 | 901.20 (-0.53% ↓) | 1526.22 | 1022.86 (-33.00% ↓) |
| 95 | 808.60 | 734.00 (-9.21% ↓) | 921.75 | 920.65 (-0.12% ↓) | 1562.25 | 1100.00 (-29.60% ↓) |
| 96 | 896.36 | 741.96 (-17.21% ↓) | 1011.32 | 934.96 (-7.56% ↓) | 1639.68 | 1121.36 (-31.61% ↓) |
| 97 | 964.64 | 938.58 (-2.71% ↓) | 2822.09 | 986.11 (-65.06% ↓) | 2930.61 | 1179.86 (-59.74% ↓) |
| 98 | 2893.70 | 1119.20 (-61.32% ↓) | 2922.12 | 1120.12 (-61.65% ↓) | 3015.38 | 1241.38 (-58.84% ↓) |
| 99 | 2928.25 | 1130.47 (-61.40% ↓) | 2928.98 | 1176.07 (-59.87% ↓) | 3037.12 | 1309.16 (-56.89% ↓) |
| 100 | 2953.00 | 1276.00 (-56.78% ↓) | 3026.00 | 1282.00 (-57.65% ↓) | 3346.00 | 1325.00 (-60.39% ↓) |

Table 4: Total Propagated Delays for 50R,50IR Scenarios with $L = (70, 80, 90)$; pth represents the pth percentile of the TPGD (Total Propagated Delays); $N$: Nominal, $R$: Reliable, with percentage change in Reliable relative to Nominal. Improvements where $R < N$ are highlighted.

| pth | 50R,50IR-70 | | 50R,50IR-80 | | 50R,50IR-90 | |
|-----|------|------|------|------|------|------|
| | N | R | N | R | N | R |
| 90 | 402.30 | 424.60 (5.55% ↑) | 530.00 | 681.40 (28.59% ↑) | 1591.70 | 1217.80 (-23.51% ↓) |
| 91 | 406.80 | 430.45 (5.82% ↑) | 558.26 | 745.93 (33.62% ↑) | 1677.56 | 1288.81 (-23.20% ↓) |
| 92 | 425.64 | 437.32 (2.75% ↑) | 595.00 | 823.28 (38.37% ↑) | 2030.44 | 1298.60 (-36.02% ↓) |
| 93 | 436.43 | 480.94 (10.19% ↑) | 871.91 | 908.82 (4.23% ↑) | 2088.97 | 1317.56 (-36.95% ↓) |
| 94 | 483.86 | 708.88 (46.51% ↑) | 884.42 | 935.94 (5.83% ↑) | 2346.04 | 1349.06 (-42.48% ↓) |
| 95 | 531.40 | 757.20 (42.43% ↑) | 891.70 | 983.40 (10.28% ↑) | 2454.50 | 1727.80 (-29.63% ↓) |
| 96 | 881.60 | 822.60 (-6.68% ↓) | 905.64 | 1011.16 (11.67% ↑) | 3037.84 | 1770.44 (-41.69% ↓) |
| 97 | 896.27 | 934.68 (4.28% ↑) | 923.85 | 1043.80 (12.99% ↑) | 3588.64 | 1979.63 (-44.82% ↓) |
| 98 | 944.76 | 994.74 (5.29% ↑) | 1054.12 | 1200.66 (13.91% ↑) | 3678.24 | 2203.68 (-40.09% ↓) |
| 99 | 2901.00 | 1283.56 (-55.74% ↓) | 2930.11 | 1290.31 (-55.96% ↓) | 3886.91 | 2678.17 (-31.08% ↓) |
| 100 | 3693.00 | 2032.00 (-45.00% ↓) | 3733.00 | 2113.00 (-43.42% ↓) | 3977.00 | 2695.00 (-32.24% ↓) |

where $\text{pgd}_i$ is the propagated delay for each flight $i$ in the pairings, $\text{SDT}_{i+1}$ is the scheduled departure time of the next flight $i + 1$, $\text{AAT}_i$ is the actual arrival time of the current flight $i$, and $\Delta$ is the minimum sit time between flights.

**Results:** We simulate both nominal and reliable solutions separately for 100 seeds each and analyze the results across the scenarios. Reliable crew pairing solutions outperform nominal solutions, particularly in scenarios with higher irregular operations and higher levels of delay. The scenarios with 75% regular operations and 25% irregular operations (75R, 25IR) at various severity levels (70th, 80th, and 90th percentiles) represent situations where delays were greater than the $pth$ percentile used to simulate delays. These scenarios are designed to test the robustness of crew pairing solutions under mixed operational conditions, highlighting how well they handle varying degrees of irregularity.

When comparing reliable and nominal solutions for 75R, 25IR scenarios in Table 3 scenarios across different severities of delay, it becomes evident that reliable solutions generally outperform nominal ones, especially at higher percentiles. For the 75R, 25IR-70 scenario, the Total Propagated Delay (TPGD) at the 99th percentile for reliable solutions is 1130.47, while for nominal solutions it is significantly higher at 2928.25. As illustrated in Figure 5, this performance gap between reliable and nominal solutions grows as the severity of delays increases, with the gap widening notably for scenarios involving delays sampled from higher percentiles (P70, P80, P90).

In the 75R, 25IR-80 scenario, the trend continues with reliable solutions showing a TPGD of 1223.89 at the 99th percentile, compared to 2998.11 for nominal solutions. Even as the severity of delays

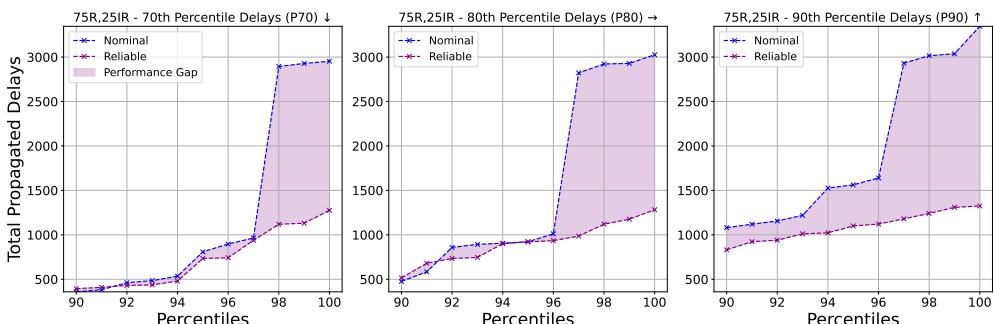

Figure 5: Performance Gap Between nominal and Reliable Solutions for 75R,25IR Across Increasing Delay Severity (P70, P80, P90). Here, 'P#': delays of Irregular runs are sampled from #-th percentile.

increases, reliable solutions maintain a lower TPGD, demonstrating their effectiveness in mitigating the impact of severity of delays. In the 75R, 25IR-90 scenario, the TPGD at the 99th percentile for reliable solutions is 1309.16 mins, whereas for nominal solutions it is much higher at 3037.12 mins. Reliable solutions consistently show better performance across these mixed operational scenarios, with a significant reduction in TPGD at higher percentiles compared to nominal solutions, showing an improvement of atleast 1000 mins (TPGD) for 98th, 99th and 100th percentiles.

As we transition from less irregular operations 100IR to more irregular operations 100IR, the magnitude of the delays increases significantly. From Table 4, we observe that for 50R,50IR, scenarios with more severity of delay shows greater performance improvements. For instance, $L = 90$ percentile, R achieves a minimum reduction of 23% across all the upper percentiles of TPGD. For scenarios 75R, 25IR-L=(70, 80, 90), 100R and 100IR, the detailed performance comparison between the reliable and nominal solutions shows similar trends of improvement and can be further summarized in Appendix G, highlighting robustness of `SurvCG` in highly irregular scenarios.

## 5 DISCUSSION

We introduce a data-driven approach for the Crew Pairing Problem (CPP) to tackle the uncertainty in real-world planning using an exposition of crew pairing in aviation operations. Our results indicate the tremendous potential to impact operational efficiencies in the real world by leveraging historical on time performance data. We accomplish this by incorporating the reliabilities, predicted using survival analysis – a popular time-to-event model, in the cost function of the CPP task and combining this with column generation algorithm – the state-of-the-art algorithm to solve CPP. `SurvCG` significantly reduces total propagated delay and deadheading connection costs compared to the nominal solution.

Reliable solutions also show a significant reduction in higher percentiles across various scenarios, demonstrating their robustness under mixed and highly irregular conditions. While recent methods Antunes et al. (2019) report a reduction of $18 - 20\%$ as compared to the nominal solution in terms of total propagated delays, we demonstrate that `SurvCG` can lead to a reduction of up to approx. 60% over nominal on this metric, that too under the challenging irregular operating conditions. For instance, in the 75R, 25IR-70 scenario in Table 3, reliable solutions save 1797.78 minutes in TPGD at the 99th percentile. Similar trends are observed in the 75R, 25IR-80 and 75R, 25IR-90 scenarios, with savings of 1955.12 minutes and 1724.70 minutes, respectively. Our solution also reduce certain costs and significantly decrease deadheading, resulting in lower operational expenses over the nominal.

`SurvCG`, is also, to the best of our knowledge, the first algorithm to incorporate data-driven reliabilities for this long-term planning problem. As a result, this investigation also lays the foundations of developing other machine learning for optimization methods. While, a limitation of our approach is that the optimal pairings obtained on solving the CPP using column generation algorithm depends on how accurate the reliability predictions are from the survival model. It is worth noting that while we use a specific survival analysis model – CoxTime, which worked well for this dataset, `SurvCG` is not constrained to using this model and practitioners can incorporate any appropriate time-to-event model. Future work on this thread can extend the use of reliabilities for other optimizations, or even combine this with recent works on reinforcement learning for column generation Chi et al. (2022). Overall, our work opens new avenues to usher operational efficiencies in the aviation industry and beyond in the backdrop of high competition, climate impacts, and customer retention.

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

## A ABBREVIATIONS AND NOTATIONS

This appendix provides a comprehensive list of the abbreviations and notations used throughout the paper.

### A.1 FLIGHT-RELATED NOTATIONS

- $\mathcal{F}$: Set of all flights
- $origin$: Origin of a flight.
- $dest$: Destination of a flight.
- SET: Scheduled Elapsed Time.
- SAT: Scheduled Arrival Time.
- SDT: Scheduled Departure Time.
- ADT: Actual Departure Time.
- AET: Actual Elapsed Time.
- AAT: Actual Arrival Time.

### A.2 COST-RELATED NOTATIONS

- $c$: nominal cost.
- $rc$: Reliable cost.

### A.3 METRICS

- $P$-index: P-index, a new metric introduced to measure the predictive power of the model.
- $C^{td}$: Total Delay Cost index.

## A.4 TIME AND SURVIVAL MODEL

- $t_q$: Query time.
- $f$: Survival model function.
- $S$: Survival function.
- $\mathrm{p}_{\equiv}$: Probability of being within the acceptable delay range.
- $\mathrm{MAE}_{>}$, $\mathrm{MAE}_{\equiv}$, $\mathrm{MAE}_{<}$: Mean Absolute Errors above, within, and below the predicted threshold.
- $\hat{q}$: Predicted time of an event.

## A.5 DATASET, NETWORK, AND CONSTRAINTS

- $\mathcal{D}$: Dataset of flights and connections.
- $\mathcal{C}$: Constraints set.
- $N$: Flight network.
- $A$: Set of arcs in the network.
- $c$: Crew base.
- $\hat{\mathcal{F}}$: Pruned set of flights.
- $\delta$: Sit or Connection time between flights (assumed to be 60 mins).

## A.6 PAIRING-RELATED NOTATIONS

- $\mathcal{P}$: Set of all pairings.
- $P_{opt}$: Set of optimal pairings.
- $p$: Single pairing.
- $\mathcal{R}$: Actual elapsed distribution.

## A.7 KDE AND RELIABILITY

- $\mathcal{KDE}$(matched flights): Kernel Density Estimation for matched flights.
- $r$: Reliability score of a flight connection.
- $\phi$: Cost function adjusted for reliability.

## B SUMMARY OF SURVIVAL MODELS

Table 5: Summary of Survival Analysis Methods

| Method | Model Type | Prop. Constraint | Main Benefit |
|---|---|---|---|
| Cox Proportional Regression Cox (1972) | Continuous | Yes | Most Interpretable |
| DeepSurv Katzman et al. (2018) | Continuous | Yes | Handles non-linearity (Uses NN for reg Cox) |
| Cox-Time Kvamme et al. (2019) | Continuous | No | Extends Cox Reg beyond prop. hazards |
| Cox-CC Kvamme et al. (2019) | Continuous | Yes | Proportional version of Cox-Time |
| Random Survival Forests Ishwaran et al. (2008b) | Continuous | No | Handles interactions and non-linearity |
| DeepHit Lee et al. (2018) | Discrete | No | Best discriminative ability (C-index) |

## C    INSTANCE GENERATION

To set up a crew pairing experiment, we first generate an instance of crew operations consisting of a closed network of flights and connections over a fixed period. The crew starts and finishes at a specified crew base. Specifically, an instance is described by a network $N$ containing a set of flights $\hat{\mathcal{F}}$, connections $\hat{A}$, and costs $\hat{C}$. Using a set of spatiotemporal constraints $\mathcal{C}$, we construct $N$ as follows:

1. Filter $\mathcal{D}$ according to $\mathcal{C}$ to obtain a set of flights $\mathcal{F}_0$.
2. Construct an initial network $N_0$ by applying space and time constraints to $\mathcal{F}_0$. A connection between flight $i$ and flight $j$ is feasible if $origin_j = dest_i$ and $\delta_{min} \leq \text{SDT}_j - \text{SAT}_i \leq \delta_{max}$, where, $\delta_{min}, \delta_{max}$ are the minimum and maximum connection times.
3. Prune $N_0$ to remove redundant flights and extract the subgraph $N$ describing our instance.

We create an instance for December 2-5, 2019, using the flight operations of Endeavor Air between all the airports in the network on these dates, with John F. Kennedy International Airport (JFK) as the crew base.

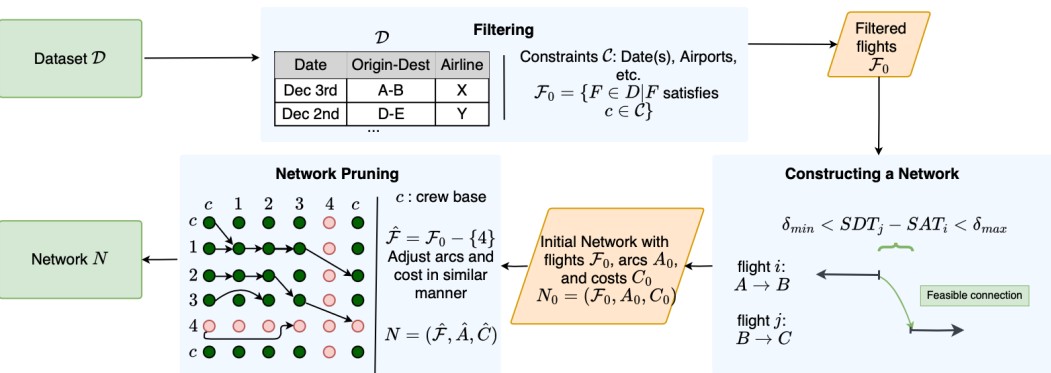

Figure 6: Block diagram for Instance Generation

**Filtering:** During the filtering phase, we specify space and time constraints to select relevant flights from the dataset. The constraints include the specific dates and the crew base for starting and ending operations. This phase aims to narrow down the vast dataset to a manageable subset that is relevant to the instance we want to create. By applying these constraints, we extract a filtered set of flights $\mathcal{F}_0$ from the dataset $\mathcal{D}$, such that $\mathcal{F}_0 = \{i \in \mathcal{D} | i \text{ satisfies } \mathcal{C}\}$. This step ensures that only flights within the specified dates and that either start or end at the crew base are included in the instance.

**Constructing a Network:** In the network construction phase, we form connections between the filtered flights. The connections represent possible pairings of flights that a crew can operate within the given constraints. The network includes nodes for the crew base and connections between flights that are feasible based on time constraints. Specifically, a connection between flight $i$ and flight $j$ is feasible if the destination of $i$ matches the origin of $j$, and the time difference between the scheduled departure time of flight $j$ and the scheduled arrival time of flight $i$ falls within the allowable connection time range. The cost for a connection is calculated as $\text{SDT}_j - \text{SAT}_i + \text{SET}_i$, where SDT is the scheduled departure time, SAT is the scheduled arrival time, and SET is the scheduled elapsed time.

**Network Pruning:** The pruning phase ensures that the network remains practical and feasible for crew pairings. During this phase, we remove redundant or infeasible flights and connections that do not contribute to viable pairings. This is done by identifying and retaining only those flights that can form a continuous path from the origin to the destination crew base. Flights that do not participate in any such path are pruned out.

## D    FLIGHT MATCHING CRITERIA AND LATE AIRCRAFT DELAY

The matched flights are identified based on the following criteria:

Table 6: Sensitivity analysis of $\alpha$. Higher $\alpha$ penalizes longer connection times. Headers: DFC - Deadhead Flying Cost, DCC - Deadhead Connection Cost, TFC - Total Flying Cost, TCC - Total Connection Cost, TC - Total Cost.

| | Deadheading Cost | | Total Cost | | |
| $\alpha$ | DFC | DCC | TFC | TCC | TC |
| --- | --- | --- | --- | --- | --- |
| 1 | 1657 | 13244 | 27942 | 216189 | 244131 |
| 0.5 | 260 (-84.30) | 1922 (-85.49) | 26545 (-5.00) | 217586 (0.65) | 244131 (0.00) |
| 2 | 2948 (77.91) | 17436 (31.62) | 29233 (4.63) | 214933 (-0.58) | 244166 (0.01) |
| 3 | 3099 (87.01) | 17130 (29.41) | 29384 (5.18) | 214777 (-0.65) | 244161 (0.01) |
| 4 | 3119 (88.17) | 17406 (31.48) | 29404 (5.23) | 214727 (-0.68) | 244131 (0.00) |
| 5 | 3119 (88.17) | 16211 (22.38) | 29404 (5.23) | 214762 (-0.66) | 244166 (0.01) |

- *Origin:* The airport from which the flight departs.

- *Destination:* The airport to which the flight arrives.

- *Time of Day of the Scheduled Arrival:* The time of day when the flight is scheduled to arrive. This can be segmented into different periods, such as morning (06:00 AM - 11:59 AM), afternoon (12:00 PM - 04:59 PM), evening (05:00 PM - 10:59 PM), and night (11:00 PM - 05:59 AM).

The *LateAircraftDelay* is not included in the initial delay estimation to avoid double-counting, as this category represents delays caused by propagation through aircraft connections. This delay is simulated separately following Antunes et al. (2019), where the delay is approximated as the difference between actual and scheduled arrival times, minus the *LateAircraftDelay*.

# E    SENSITIVITY ANALYSIS OF $\alpha$

**Sensitivity analysis of $\alpha$:** For $\alpha = 0.5$, deadhead fly and connection costs drop significantly (-84.3% and -85.49%, respectively), with only a slight increase in total connection cost (+0.65%) as shown in Table 6. As $\alpha$ increases to 1, costs rise, with deadhead fly cost increasing by 77.91% and connection cost by 31.62%. At $\alpha = 3$, deadhead fly cost peaks (+87.01%), with minimal changes for higher values. Deadheads increase from 2 at $\alpha = 0.5$ to 28 at $\alpha \geq 3$, indicating diminishing returns beyond this point.

# F    DETAILED SCENARIO NOTATIONS

Table 7: Simulation Scenarios Based on Percentage Irregularity and Level of Delay. R: Regular Operations, IR: Irregular Operations. For IR runs, delay values are sampled from the specified percentile using KDE.

| Scenario Notation | Description |
| --- | --- |
| 100R 0 | 100% of runs are R |
| 75R, 25IR 70 | 75% of runs are R, 25% from >70 percentile |
| 75R, 25IR 80 | 75% of runs are R, 25% from >80 percentile |
| 75R, 25IR 90 | 75% of runs are R, 25% from >90 percentile |
| 50R, 50IR 70 | 50% of runs are R, 50% from >70 percentile |
| 50R, 50IR 80 | 50% of runs are R, 50% from >80 percentile |
| 50R, 50IR 90 | 50% of runs are R, 50% from >90 percentile |
| 25R, 75IR 70 | 25% of runs are R, 75% from >70 percentile |
| 25R, 75IR 80 | 25% of runs are R, 75% from >80 percentile |
| 25R, 75IR 90 | 25% of runs are R, 75% from >90 percentile |
| 100IR 70 | 100% sample delay from >70 percentile |

# G DETAILED PERFORMANCE COMPARISON

As we analyze the results, Figure 7 clearly shows the trends in performance as irregular operations increase. The total propagated delays (TPGD) become more severe as both irregularity levels and the percentiles of delay rise, demonstrating the greater importance of incorporating reliability into decision-making, especially when met with disruptions.

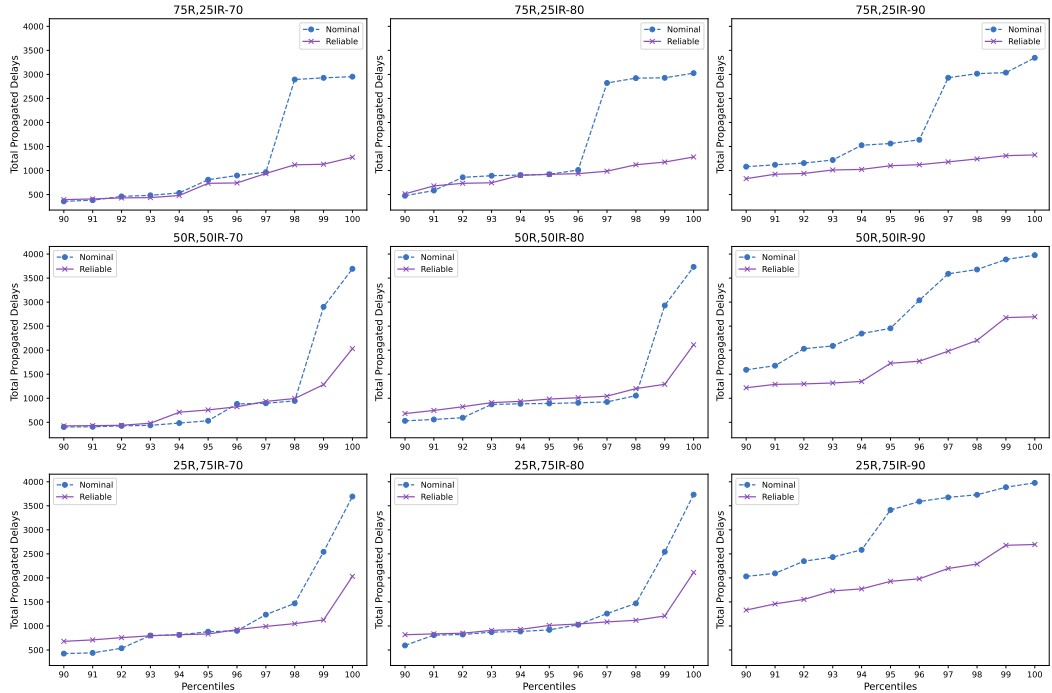

Figure 7: Total Propagated Delays for 75R,25IR Scenarios (70, 80, 90). N: Nominal, R: Reliable. The "75R,25IR" denotes the percentage of regular and irregular runs, respectively. The numbers 70, 80, and 90 indicate the level of delay in each scenario, which increases from left to right. Total Propagated Delays show significant improvements as irregularity increases or as the level of delay rises.

Table 8: Total Propagated Delays for 25R, 75IR Scenarios with $L = (70, 80, 90)$; pth represents the pth percentile of the TPGD (Total Propagated Delays); $N$: Nominal, $R$: Reliable, with percentage change in Reliable relative to Nominal. Improvements where $R < N$ are highlighted.

| pth | 25R, 75IR-70 | | 25R, 75IR-80 | | 25R, 75IR-90 | |
|---|---|---|---|---|---|---|
| | **N** | **R** | **N** | **R** | **N** | **R** |
| 90 | 425.80 | 679.90 (59.68% ↑) | 594.80 | 817.80 (37.49% ↑) | 2031.30 | 1330.50 (-34.50% ↓) |
| 91 | 440.20 | 710.32 (61.36% ↑) | 809.81 | 834.99 (3.11% ↑) | 2094.39 | 1459.65 (-30.31% ↓) |
| 92 | 536.04 | 757.20 (41.26% ↑) | 822.24 | 849.96 (3.37% ↑) | 2347.72 | 1552.12 (-33.89% ↓) |
| 93 | 801.56 | 795.68 (-0.73% ↓) | 871.91 | 908.05 (4.14% ↑) | 2432.42 | 1728.52 (-28.94% ↓) |
| 94 | 813.32 | 818.60 (0.65% ↑) | 885.80 | 927.28 (4.68% ↑) | 2583.38 | 1928.35 (-31.41% ↓) |
| 95 | 881.20 | 832.70 (-5.50% ↓) | 919.10 | 1011.45 (10.05% ↑) | 3413.10 | 1981.84 (-42.00% ↓) |
| 96 | 898.76 | 924.68 (2.88% ↑) | 1025.44 | 1040.80 (1.50% ↑) | 3589.52 | 2196.58 (-44.79% ↓) |
| 97 | 1235.57 | 990.74 (-19.82% ↓) | 1257.88 | 1084.96 (6.51% ↑) | 3675.56 | 2287.96 (-37.79% ↓) |
| 98 | 1469.66 | 1117.66 (-24.03% ↓) | 1469.66 | 1117.66 (10.06% ↑) | 3729.20 | 2678.17 (-28.09% ↓) |
| 99 | 2542.62 | 1125.16 (-55.75% ↓) | 2543.02 | 1208.14 (-52.42% ↓) | 3886.91 | 2695.00 (-30.45% ↓) |
| 100 | 3693.00 | 2032.00 (-45.00% ↓) | 3733.00 | 2113.00 (-43.42% ↓) | 3977.00 | 2695.00 (-32.24% ↓) |

Table 9: Total Propagated Delays for $100R - 0$ and $100IR - 70$ scenarios; pth represents the pth percentile of the TPGD (Total Propagated Delays); $N$: Nominal, $R$: Reliable, with percentage change in Reliable relative to Nominal. Improvements where $R < N$ are highlighted.

| pth | 100R-0 | | 100IR-70 | |
|-----|--------|--------|----------|--------|
| | **N** | **R** | **N** | **R** |
| 90 | 237.40 | 253.00 (6.57% ↑) | 801.30 | 710.80 (**-11.30%** ↓) |
| 91 | 241.00 | 266.41 (10.55% ↑) | 804.45 | 757.60 (**-5.83%** ↓) |
| 92 | 245.72 | 312.28 (27.09% ↑) | 814.76 | 795.92 (**-2.31%** ↓) |
| 93 | 300.77 | 328.47 (9.22% ↑) | 881.28 | 818.70 (**-7.10%** ↓) |
| 94 | 313.76 | 351.54 (12.05% ↑) | 885.66 | 837.66 (**-5.42%** ↓) |
| 95 | 359.20 | 423.35 (17.86% ↑) | 912.65 | 991.90 (8.67% ↑) |
| 96 | 403.36 | 734.00 (82.01% ↑) | 1237.76 | 1049.76 (**-15.18%** ↓) |
| 97 | 473.35 | 739.64 (56.24% ↑) | 1491.35 | 1116.39 (**-25.12%** ↓) |
| 98 | 937.52 | 922.22 (**-1.63%** ↓) | 2894.20 | 1131.94 (**-60.88%** ↓) |
| 99 | 2534.97 | 934.86 (**-63.12%** ↓) | 2960.40 | 1283.56 (**-56.64%** ↓) |
| 100 | 2928.00 | 1119.00 (**-61.78%** ↓) | 3693.00 | 2032.00 (**-45.00%** ↓) |

