# OpenReview forum: "Uncertainty Aware Column Generation for Crew Pairing Optimization Using Survival Analysis"
_ICLR.cc/2025/Conference — Submitted to ICLR 2025_

### Official Review · Reviewer_yQi4 · 2024-10-27

**Soundness:** 2
**Presentation:** 2
**Contribution:** 1
**Rating:** 1
**Confidence:** 5

**Summary:**

The paper studies a crew scheduling problem in airline industry and proposes a column-generation based algorithm to dynamically quantify uncertainties using operational patterns in historical data.

**Strengths:**

The paper focuses on an NP-hard problem and considers uncertainties in crew scheduling, and aim to improve reliability for real-world problems.

**Weaknesses:**

1.	Crew scheduling is a classical problem in airline operations and is a well studied topic. I do not see new concepts or technical modeling/algorithmic challenges brought up by this paper.
2.	Considering uncertainties in airline crew scheduling are not new, and using data-driven approaches for scheduling are not new either. The contribution of this study is very limited.
3.	The paper only considers crew scheduling but assumes that flight schedules, flight assignment (to routes) and their gate assignments are given. This further limits the complexity of the problem and it is not considered as state-of-the-art in airline operations research area.
4.	The column generation (CG) based algorithm is very standard and with another layer of “survival analysis” it does not seem to add significant changes or difficulty to use CG for crew scheduling. I do not see efforts being made to improve the scalability and computational complexity of CG, which could suffer from the combinatorial structure of the paper.
5.	The proposed methods in this paper are weakly related to the focus of this conference.
6.	I do not see any data-driven type of theoretical analysis although the authors state that they are the first to propose “data-driven solutions for uncertainty-aware reliable scheduling”. That is, how the data and sample sizes can affect the solution results and quality?

**Questions:**

1.	What are learning based methods studied in this paper? How this work is related to this conference?
2.	How are the approaches proposed in this work compared to the state-of-the-art integer programming and stochastic programming based methods used for solving scheduling problems in Operations Research area?
3.	What is the data-driven aspect of this approach?
4.	What are the dynamic uncertainty quantification procedures used in the paper? All seem to be static except that the data and models are connected iteratively in a rolling-horizon way. This cannot be called “dynamic”.

---

> ### Author Response · Authors · 2024-12-01
> **Response to Reviewer yQi4**
>
> We thank the reviewer for providing us with the opportunity to discuss the broader context in which SurvCG makes real-world impact.
>
> - *Paper’s fit to ICLR*: ICLR prides itself on being an inclusive venue to bring together multidisciplinary perspectives on machine learning, including topics such as uncertainty quantification, issues regarding large-scale learning and non-convex optimization, and their applications in sustainability and planning [1]. In this regard, our paper accomplishes state-of-the-art results on a real-world planning task – flight crew pairing optimization -- in a data-driven way by uncertainty quantification using survival analysis. This not only aligns with the conference’s focus, it also addresses an urgent need that plagues the future of aviation industry– dwindling profit margins in the backdrop of climate change. Therefore, we hope this provides additional perspective for the reviewer’s consideration.
>
> - *Novelty*: Recent works on crew scheduling (Lu et al. 2015), including the current state-of the art (Antunes et al. 2019) are based on robust optimization, wherein they plan for the worst-case delays based on historical delays or predefined uncertainty sets. This leads to very conservative planning, since delays distributions are often long-tailed. For instance, the best results that Antunes et al. 2019 achieves is ~23% reduction in delays over nominal at 94th percentile, and an average reduction in delays of ~18 to 19%. We see a maximum reduction of 61%, and ~41% decrease in delays on an average. This shows that a machine learning-based approach to tackle this classical operations research task can significantly impact real world planning.
>
> Next, unlike the previous work we do not just use statistics based on historical data, our survival analysis component is a neural network model which is inherently data-driven.  Not only does this survival analysis model provide custom forecasts for a flight (a first in the literature), it also leads to unprecedented efficiencies (Tab 3, 4) as compared to these methods because of reliabilities.  Next, the modular nature of our framework means that the survival analysis component can be used with other optimizations for various stages of aviation planning. Moreover, since survival analysis model here learns the underlying representations, mapping the features for a specific flight to a time-to-event, it can also be used for new flights (flights between an origin and a destination at a new time for which historical data does not exist), where existing uncertainty set-based methods cannot. For instance, a flight between an origin and destination at a different time can have different delay characteristics. Furthermore, due to this data-driven nature of our method, it is “dynamic” in the sense that when a given schedule changes, our reliabilities can also change while this is not true for the previous works since they use the same uncertainty sets. This is what we refer to as “dynamic” since it is based on individual flights and times. We would like to apologize if this caused any confusion. However, we would like to politely disagree with the initial comments from the reviewer, and we hope this provides additional context of the recent works in the area.
>
> - *Flights schedules are assumed to be known at the time of crew-pairing*: We would like to gently correct the reviewer’s understanding: assuming flight schedules when solving crew pairing problem is the standard way to solve this real-world large-scale optimization in the industry as well as in the academic literature (see Antunes et al. 2019 and the references therein). This is an especially important problem to study since even when the schedules are known, there are numerous ways to pair the crew with the flight, and the uncertainty induced by delays can break a pairing easily. Therefore, we would like to urge them to not reject the paper based on this misinterpretation of the task.
>
> We hope this helps to re-evaluate our work positively, and we will be happy to clarify any other aspects. Thank you again.

---

> > ### Author Response · Authors · 2024-12-03
> > **Official comment by authors**
> >
> > Dear Reviewer,
> >
> > Thank you for your valuable feedback on our paper. We've incorporated your suggestions into the revised rebuttal and would appreciate your thoughts on the changes and clarification. Please let us know if you need any further questions or have additional concerns. Hope this leads to a positive evaluation of our work.
> >
> > Thank you for your time and consideration.

---

### Official Review · Reviewer_iZff · 2024-11-03

**Soundness:** 3
**Presentation:** 3
**Contribution:** 3
**Rating:** 6
**Confidence:** 3

**Summary:**

This paper presents SurvCG, a novel approach to the Crew Pairing Problem (CPP) in airline operations that integrates survival analysis into a column generation framework to account for uncertainties in flight operations. Traditional CPP formulations are deterministic and often fail to consider real-world disruptions like delays and crew absenteeism. SurvCG addresses this by predicting flight connection reliability using survival analysis models and incorporating these reliability estimates into the cost function of the CPP. The method is evaluated on a large-scale, real-world dataset from Endeavor Air's 2019 operations under various scenarios, including regular and irregular operating conditions with varying levels of delay severity. Experimental results demonstrate that SurvCG can significantly reduce total propagated delays—by up to 61% compared to baseline methods.

**Strengths:**

- The proposed method effectively combines survival analysis for predicting flight connection reliability with the column generation method. This integration allows the optimization process to account for uncertainties in a data-driven manner.
- Experimental results show that SurvCG can reduce total propagated delays by up to 61% compared to nominal solutions, particularly under irregular operating conditions with high delay severities (e.g., the 75R scenario) highlighting the practical effectiveness of the proposed approach.
- The authors use a comprehensive real-world dataset consisting of 97,294 flights from Endeavor Air's 2019 operations. They conduct extensive simulations across various scenarios by varying the percentage of irregular operations and delay severities. The evaluation is thorough and grounded in real operational settings.

**Weaknesses:**

- While the paper references existing stochastic and robust optimization approaches for CPP, it does not seem to provide direct experimental comparisons with these methods. Without such a comparison, it is challenging to assess how SurvCG's performance stacks up against state-of-the-art robust optimization techniques under similar conditions.
- The effectiveness of SurvCG heavily relies on the accuracy of the survival analysis model used for predicting flight connection reliability. The paper does not explore how errors or uncertainties in reliability predictions might impact the overall optimization results and whether the method remains robust under less accurate predictions.
- The simulation of delays may not capture the full complexity of delay propagation in airline networks, such as cascading effects and interactions between flights. This simplification could affect the validity of the simulation results and the observed performance improvements.
- The paper does not report computational times or discuss the scalability of SurvCG in terms of solution time compared to the nominal approach. I think this information is crucial for practical implementation in real-world airline scheduling systems where time constraints are significant.

**Questions:**

In addition to the points mentioned in previous sections, some additional questions/suggestions:

- To strengthen the contribution, it would be valuable to include a direct experimental comparison of SurvCG with existing robust or stochastic CPP methods.
- Include information on the computational time and resources required by SurvCG compared to baselines. Discuss any additional overhead introduced by the proposed approach and whether this overhead is acceptable in practical settings.
- To demonstrate the generalizability of SurvCG, it would be beneficial to evaluate the method on datasets from multiple airlines, different regions, or more recent time periods. Of course, I imagine the data may not be readily available for these cases - if that’s the case feel free to ignore this suggestion.

---

> ### Author Response · Authors · 2024-12-01
> **Response to Reviewer iZff (Part 1)**
>
> We sincerely appreciate the reviewer's thoughtful comments and valuable insights, which have provided us with an opportunity to clarify and strengthen key aspects of our work.
>
> - *Direct experimental comparisons to other methods*: Unfortunately, comparing to previous research such as Schaefer et al. 2005, Lu et al. 2015, and Antunes et al. 2019 is a prohibitive task for two main reasons. First, the instances are either not provided/available or they are not sufficient to carry out the reliability since we do learning which requires historical data for learning and testing as well simulation later on.  Second, the codes are not available and implementing them is a significant undertaking. Unfortunately, there is a lack of common test instances in this context. Our work is the first to develop and provide an instance generation procedure from public available BTS data. This allows for such comparisons in the future.  From that perspective, our work is the only reproducible work as we provide pseudocode and all function forms to generate instances, set up the optimization problems, solve them, and simulate.  It is common is such type of work to compare to the deterministic baseline. Our testing has been discussed and validated with our industry partner. Nevertheless, from their paper, the best results that Antunes et al. 2019 achieves is ~23% reduction in delays over nominal at 94th percentile, and an average reduction in delays of ~19%. We see a maximum reduction of 61%, and ~41% decrease in delays on an average. This shows that a machine learning-based approach to tackle this classical operations research task can significantly impact real world planning.
>
> - *Rigorous evaluation of the effectiveness of survival models*: The reviewer is absolutely correct, the accuracy of the survival models is key in this application where the exact time of landing matters. We would like to gently correct the reviewer’s understanding that the paper does not show results to demonstrate this aspect. In fact, one of the main contributions of our work is to propose a new survival analysis metric – the P-index – which measures the accuracy of predicted survival curve to the underlying Kaplan Meier estimator (the closer to 1 the better). This is because the exact time of landing is crucial, but existing survival analysis metrics are just based on ordering to measure the model’s capability to predict which event happens before the other. We perform extensive evaluations in Table 1 using Endeavor Air’s 2019 data (97294 records). Specifically, we perform extensive experimentation using three learning rates [0.0001,0.001,0.01], three batch sizes [16,32,64], and Monte Carlo simulations with 3 random seeds for each setting. We evaluate models using metrics such as Concordance, Within, Within_pct, MAE, and others. This demonstrates that the CoxTime model not only performs well on traditional metrics like C-index, but also on P-index. Moreover, it is important to note that while our metric P-index will be key to drive advancements in the use of time-to-event models for real-world optimization tasks, our framework is not tied to choice of this survival model itself, and any performant model can be integrated into the proposed framework.
>
> - *“simulation of delays may not capture the full complexity of delay propagation”*: Because reliability is derived from real data, which inherently captures the system dependencies. Moreover, our simulation methodology is aligned with how other works in the literature validate their method (Antunes et al. 2019). Specifically, since delay propagation impacts the departure and arrival times of flights which impacts the reliability of a connection. So, the delays are captured in the data, which is used to calculate reliabilities. In the introduction and related work sections, we review methods that attempt to model and delay and discuss how our work addresses the shortcomings. We in fact focus on reliability of a connection between two flights when performed by the same crew, which would take into account delays, but also other system features like effects of congestion.
>
> - *Computational time and practicality*: Thank you for this observation, for Reliable 1 ($\lambda_1$ = 10, $\lambda_2$ = 3), and Reliable ($\lambda_1$ = 20, $\lambda_2$ =4) CG Execution Times are 1426.88 seconds and 1412.83 respectively. While this metric for Nominal is 750.48 seconds. We have added a statement regarding this on lines 337-339 in the revision. In terms of overhead, this execution time is less than 30 minutes, and since this planning is undertaken 4–5 months in advance, it is perfectly acceptable for real-world planning.

---

> ### Author Response · Authors · 2024-12-01
> **Response to Reviewer iZff (Part 2)**
>
> - *Evaluation on multiple dataset*: Thank you for this suggestion. We chose to conduct this analysis on Endeavor since our industry partners have knowledge about these airlines’ operations due to their commercial relationship. This helped us immensely to develop the method and analyze the results. Indeed, this is an extremely time-consuming task, but our method is not airline dependent, and while it would be great to run the method on other dataset, it will involve data cleaning steps which means that this is not readily available at this time.
>
> We are grateful for the reviewer's careful consideration of our work and hope that our responses have addressed their concerns, which can lead to a more positive evaluation of our work. We are happy to answer any further questions.

---

> > ### Author Response · Authors · 2024-12-03
> > **Official comment by authors**
> >
> > Dear Reviewer,
> >
> > Thank you for your valuable feedback on our paper. We've incorporated your suggestions into the revised rebuttal and would appreciate your thoughts on the changes and clarification. Please let us know if you need any further questions or have additional concerns. Hope this leads to a further positive evaluation of our work.
> >
> > Thank you for your time and consideration.

---

### Official Review · Reviewer_qSbi · 2024-11-03

**Soundness:** 2
**Presentation:** 2
**Contribution:** 2
**Rating:** 5
**Confidence:** 4

**Summary:**

The paper presents an approach for dealing with flight delays in the context of airline crew pairing optimization. In particular, it proposes to add a penalty term to flight connections that relates to the risk of missing the connection in the pricing subproblem of a column generation-based approach to solving the crew pairing problem. The penalty term is computed based on a prediction that is made by a survival analysis model that is trained on historical flight dely data.

In a set of computational experiments with one instance, first the cost structure of the solution obtained with the proposed (reliable) model is compared to a those obtained with a nominal model. Then, the results from simulations that propagate the delay are compared for both solutions under variouse levels of irregularity. It turns out that the solutions obtained with the reliable model show a much smaller total progagated delay, in particular under heavily irregular conditions.

**Strengths:**

To the best of my knowlegde, the idea of using survival analyis for estimating missed crew connections in the context of airline crew scheduling is new and original.

Also, the idea of incorporating the risk of missing a connection via a penalty term in the pairing generation subproblem is meaningful, but at the same time often used in practice and in more sophisticated ways in the literature (e.g. in Schaefer et. al. 2005, Transportation Science).

**Weaknesses:**

The authors do not describe their crew pairing problem very well. The crew pairing problem is a very characterized by both a very complex set of regulations (e.g. rules related to total block time per day, total work time per day, rest requirements between duties) and rules and by a very complex cost structure, both of which are at least in parts very airline-specific. The authors neither describe which rules and regulations they consider. Regarding the cost of a pairing, it appears that the costs of a pairing additively depends on the time of the flights and on the (weighted) connection times. This is a relatively simplistic assumption since in the real world, the cost structure of a pairing is much more complex, see e.g. the paper (Atnunes et al. 2019) that was cited in the manuscript, where the cost relates to the maximum of three different values, including the guaranteed crew pay per duty.

The authors write that they use a column-generation-based solver to solve a set covering formulation of the CPP. Column generation, however, can only be used to solve the LP relaxation of this formulation. In order to solve the problem to optimality (as claimed in the paper), they need use a so-called branch-and-price algorithm which involves a lot of design decisions, and also the use of commercial LP and/or MIP solvers. This aspect, however, is not even mentioned in the paper. Also, in the experimental results section, this is not discussed, which means that the results are not reproducible at all. The paper does not even mention how long it takes to solve the problem.

The authors vaguely write that the subproblem they solve is a shortest path problem. Without further details, one would assume that we deal with a standard shortest path problem that is easy to solve. In realistic crew pairing settings, however, the subproblem is an NP-hard resource-constrained shortest path problem. Again, this aspect should be clarified.

It is not described in detail how the value r_ij enters the optimization process. While Fig. 1 would let us assume that the prediction model is only queried when needed, that does not make too much sense to me since it should be possible to compute all r_ij values for connections in advance.

The paper seems to completely ignore the fact that the minimum connection time for crews between two flights depends on the question whether the crew remains on the same aircraft or not. Usually, if the crew stays on the same aircraft, there is no minimum connection time at all. In particular, if a flight is delayed and the crew stays on the same aircraft, there is no risk of missing the connection (which is why airlines tend to disencourage aircraft changes of crews during a duty). This aspect should be definitely be discussed in the paper, since neglecting this aspect renders the whole analysis much less useful.

The paper only focuses on flight delays, and when solving the column generation subproblem, delay propagation is not considered at all. In addition, the paper neglects complex interactions between delays and crew regulations (e.g. a large total delay may render a duty invalid) and costs (which may involve so-called time away from base). For an example how a data-driven approach to crew pairing can account for these aspects, see e.g. the Transportation Science paper (Schaefer et. al. 2005)  "Airline Crew Scheduling Under Uncertainty".

Regarding the experiments, all results are given for a single instance, and even that instance is not described very well. Only from the text I can try to infer that it deals with 243 flights, which, for real-world crew pairing is a  small-to-medium-sized problem. As mentioned above, other important aspects such as the cost structure of a pairing are not discussed.

I find that the choice of alpha=2 for the "nominal problem" is somewhat arbitrary since choosing alpha>1 means that short (and thus non-robust) connections are favored. In practice, most airlines will use some way of penalizing high-risk connections, either relying on simple rules (that are certainly more complex that just a simple factor of the connection time), or on some data-driven quantitative model. If would have been interesting to see how the survival-based approach compares to a more standard ML approach for determining delay risk.

**Questions:**

Please provide a description of the rule set and cost structure you used in your approach. Is my assessment right that you only use time-based costs for valuating a pairing?

 Is the pricing problem a (pure) shortest path problem, or a resource-constrained shortest path problem?  What is the algorithm you use for solving the pricing subproblem?

Please provide a more detailed description of the solution approach you use. How does the branch and price work?
In the experimental results, please tell us more about the hardware you run the experiments on, the solvers you use (if any), and regarding the solution time required; also if applicable the optimality gaps.

Is there any reason to query the delay model during the solution of the CG subproblem instead of computing all r_ij values a priori?

How does your model account for (or could be extended to account for) different connection time requirements based on whether the crew changes aircraft?

Please provide a more detailed description of the instance: How many flights does it involve? Do you consider only a single crew base?

I encourage you to run experiments with more instances to strengthen the evaluation of your approach.

In the appendix, you perform a sensitivity analysis regarding the total cost depending on alpha. For me, it would be very interesting to see how the total propagated delay in the simulation looks like for nominal solutions computed with different values of alpha.

Also, I would like to encourage you to test your approach against additional baselines, in particular to other approaches for choosing penality costs for risky connections.

---

> ### Author Response · Authors · 2024-12-01
> **Response to qSbi (Part 1)**
>
> We are deeply grateful to the reviewer for their exceptionally thorough and insightful evaluation of our manuscript. The level of detail and careful consideration evident in their comments demonstrates a significant investment of time and expertise, which we greatly appreciate.
>
> - *Differences from Schaefer et al.* : We thank the reviewer for suggesting this reference. Schaefer et al. (2005)  attempted to address the issue of mismatch between the crew cost used at the planning stage and the real crew cost that is based on the day of operation. They do this by simulating a crew pairing solution and estimating the operational crew cost based on historical data.  This is a good evaluation tool for a particular solution and does not answer the question of how to build a solution that will reflect operational cost at the planning stage. To do that, they suggest penalizing undesirable features of a pairing but use them in a post-optimization step as an improvement to the nominal solution, i.e., they modify the nominal solution to improve it with respect to the penalties. This is fundamentally different from our framework, where we incorporate the reliability of a connection in the pairing at the planning stage.  Optimizing for reliability in addition to nominal cost at the planning stage leads to solutions that are less likely to be affected by delays, which in turn leads to the planning cost to better reflect operational cost.  The reference is discussed on page 2 lines 75-76 and page 3 lines 111-113.
>
> - *Incorporating Additional Rules in our Crew Pairing Task*:  The rules are those that can be incorporated in a flight network and similar to the ones modelled in the literature ((Schaefer et al. 2005, Lu et al. 2015 Atunes et al. 2019). It is true that there are other complex rules on the feasibility of a duty and a pairing, so the general practice is to apply those when possible at the modelling stage, at the duty and pairing generation stage and alternatively when finding a feasible solution from the generated columns.  Since the rules we model are standard in the literature, we omit the description due to the limited number of pages.
>
> - *Justification of the Cost Function*: In the CG framework, similar to other works in the literature, the cost function is linear (sum of $c_p$ $x_p$). However, the pairing cost $c_p$ may be calculated using a nonlinear function. Schaefer and Atunes use the pay-and-credit model, and we use a penalty function. Our choice of the penalty cost function was verified by our industry partner as aligned with their practice. We can easily switch to a pay-and-credit model, and the whole framework will not change. A clarification is added on page 6, lines 303-305.
>
> - *Justification for using CG*:  It is true that CG only solve the master problem and does not provide a feasible solution. However, it is well known that for these types of problems, the LP bound is very tight. Hence, to obtain a near-optimal solution, it is sufficient to add the integer requirement and solve the CG master problem as an IP. As noted, the optimality gap was about 0.05\%, which makes implementing branch-and-price not worth it.  What we do to obtain a near-optimal solution is common practice in this domain. All the results are reproducible, and our implementation may solve any crew pairing instance. We discuss details about the solutions in the original submission on page 7, lines 334-337, and we have added a clarification about computational times and solver version in lines 337-339.
>
> - *Shortest path problem in crew-pairing*: We use the shortest path on a directed acyclic graph, with no resource constraints. Even though the costs on the arcs of the shortest path problem (SPP) may be negative within the CG algorithm, the graph underlying the SPP is directed acyclic (does not have cycles) because the flight network is a time-space network.  Also, there are no resource constraints. Hence, the SPP remains easy to solve (P not NP) and a label setting algorithm applies because there are no directed cycles.  The SPP with resource constraints appears in CG for the vehicle routing problem with time windows, not for crew pairing optimization.
>
> - *Role of $r_{ij}$ in Fig. 1*:  The connection between $r_{ij}$ and the optimization in Fig. 1 is just used to indicate how these scores enter the optimization. For this exposition, we precompute these values. However, the overall framework is not tied to this optimization, which communicates the potential of future optimization procedures which may query $r_{ij}$’s on-the-fly as needed.  We have added this clarification in Fig. 1 and in the description.

---

> ### Author Response · Authors · 2024-12-01
> **Response to qSbi (Part 2)**
>
> - *“minimum connection time for crews between two flights depends on whether the crew remains on the same aircraft or not.”*: Here, we assume a push-back recovery strategy similar to the literature (Schaefer et al. 2005, Lu et al. 2015 Antunes et al. 2019). In this policy, delay is propagated by pushing the departure time of subsequent flights when needed. It is a commonly used in airlines in addition to other more extreme and more expensive policies like flight cancellations. It is reasonable to use a push-back policy at the planning stage, as the purpose is to come up with a crew pairing solution that can be recovered by push-back and not by other recovery policies. Incorporating these policies are interesting future research directions. The push-back policy assumption is clarified in the manuscript on page 6, lines 310-311.
>
> - *“delay propagation is not considered at all”*: Delay propagation impacts the departure and arrival times of flights, which impacts the reliability of a connection. So, the delays are captured in the data, which is used to calculate reliabilities. In the introduction and related work sections, we review methods that attempt to model and delay and discuss how our work addresses the shortcomings. We in fact focus on reliability of a connection between two flights when performed by the same crew, which would take into account delays, but also other system features like effects of congestion. Moreover, our simulation methodology is aligned with how other works in the literature validate their method (Antunes et al. 2019).
>
> - *Use of a single instance*: Appendix C describes our procedure to generate instances. And the whole framework including the instance generation and simulation parts are detailed and may be reproduced. While we excluded some details in the main paper due to the limit on the number of pages, we aim to release the instance upon acceptance with additional details, and have provided the instance as a supplementary. For context, the most recent work of Antunes et al. 2019 uses a single instance with 94 flights. Schaefer et al. 2005 does not provide any detail about their data. Therefore, our framework is superior both in terms of the scale as well as the public availability of our instance.
>
> - *Choice of $\alpha=2$*: This choice was made based on the consultations with our industry partners, where they asked us to penalize the sit time of the crew to avoid minimize the time between flights (for a crew). Using the penalty function, one can generate multiple solutions with different cost and reliability values to provide choices for the decision maker, who may use their expert opinion to favor different solutions for different settings. Also, our framework is modular and reliability may be integrated within other cost functions and with other union or airline specific rules.
>
> - *Justification for use of survival analysis instead of classical regression ML models*: This is a very nuanced point – thank you for this observation. The main reason why we opted for survival analysis for this task is that while ML models (say a regression model) may predict the delay for a flight, it does not give an indication of what percentage of flights will achieve this. In fact, since this is just an average, it cannot be readily used in an optimization and not precise enough for planning. On the other hand, since survival analysis predicts a survival function, it can inform us what percentage of flights are expected to land at a given time (see Fig 2). This not only gives a precise estimate of expected behavior, and a way to choose a reliability threshold, this reliability can be readily used in an optimization.
>
> - *“How does your model account for different connection time requirements based on whether the crew changes aircraft?”* This can be easily incorporated in calculating the reliability. Specifically, as shown in Fig. 2, the survival function predicted by the survival analysis model can be used to select a different connection time.

---

> ### Author Response · Authors · 2024-12-01
> **Response to qSbi (Part 3)**
>
> - *Sensitivity Analysis*: Since the nominal cost, $c_{ij}$ is calculated based on alpha and used as is in the nominal problem, which they aim to lead to quality nominal crew pairing solutions, $c_{ij}$ is then used to calculate the reliable cost $rc_{ij}$. Changing $\alpha$ is expected to change both the nominal and reliable solutions. An interesting future research direction is to focus on modelling other crew pairing cost structures and comparing with our cost function for different values of alpha. This, however, will not change the importance of adding reliability of a connection to any pay function. Referring back to the literature, e.g., Antunes et al. 2019, the pay-and-credit cost function also requires a number of parameters that are generally airline specific, not advertised, and very hard to determine. They all select one parameter setting that they do all the testing on. This is not ideal and emphasizes the need for a more comprehensive study on crew pay structures, and that is a research direction we plan to pursue in the future.
>
> - *Comparisons to other works*: Unfortunately, comparing to previous research such as Schaefer et al. 2005, Lu et al. 2015, and Antunes et al. 2019 is a prohibitive task for two main reasons. First, the instances are either not provided/available or they dare not sufficient to carry out the reliability since we do learning which requires historical data for learning and testing as well simulation later on.  Second, the codes are not available and implementing them is a significant undertaking. Unfortunately, there is a lack of common test instances in this context. Our work is the first to develop and provide an instance generation procedure from public available BTS data. This allows for such comparisons in the future.  From that perspective, our work is the only reproducible work as we provide pseudocodes and all function forms to generate instances, set up the optimization problems, solve them, and simulate.  It is common is such type of work to compare to the deterministic baseline. Our testing has been discussed and validated with our industry partner.
>
> We sincerely thank the reviewer again for their invaluable input. We hope these responses lead to a positive evaluation of our work. We are more than happy to address any further questions or concerns they may have.

---

> > ### Comment · Reviewer_qSbi · 2024-12-03
> >
> > Thank you for your detailed (but unfortunately very late) response, which clarifies many of the issues I raised, and also for improving the paper.
> >
> > However, I find that the paper still lacks clarity with respect to some very important aspects, and some of my suggestions were not addressed. For example, I did not ask the authors to conduct new experiments with existing approaches, but it would have been really interesting to compare the performance of a standard regression approach (or, e.g. a quantile regression approach) for predicting delays in the same framework.
> >
> > Also, I asked the authors to include results showing the total propagated delays for values of alpha different from 2, this did not happen, although it would have been very relevant.
> >
> > All in all, I increase my evaluation of the paper, but I still consider think that it has certain weaknesses to be addressed.

---

### Official Review · Reviewer_ps4T · 2024-11-09

**Soundness:** 3
**Presentation:** 3
**Contribution:** 3
**Rating:** 5
**Confidence:** 3

**Summary:**

This study propose SurvCG, a reliability-focused approach to the crew pairing problem (CPP) that incorporates survival analysis and column generation to handle real-world uncertainties. By forecasting flight connection reliability, SurvCG significantly reduces propagated delays, achieving up to 61% improvement over traditional models.

**Strengths:**

- Survival analysis allows SurvCG to predict and include the reliability of flight connections, offering a more realistic and robust solution for crew scheduling.
- The method used here is an existing one, but there are new regulations for the framework to deal with flight schedule delays.

**Weaknesses:**

- The evaluation of the calculation time for the main problem is feasible if it is relatively small, but the scalability evaluation is insufficient.
- Flights not only get delayed, but also arrive earlier than expected or get canceled. It is only optimized for a part of the process.
- The performance evaluation and explanation of the survival model are insufficient, and the data set is also limited.

**Questions:**

- Theoretically, Could you estimate the upper and lower limits of safety evaluation?

- Please evaluate the calculation amount for the master problem and clarify the size of the current situation. If this part is challenging to execute, the frame itself will not be able to operate.

- Please evaluate the details of the survival model and the impact of the model's performance.

---

> ### Author Response · Authors · 2024-12-01
> **Response to Reviewer ps4T**
>
> We sincerely appreciate the reviewer's thoughtful comments and the opportunity to clarify and expand upon the key aspects of our work.
>
> - *Computational Time and Scalability*: Thank you for this question. For Reliable 1 ($λ_1$ = 10, $λ_2$ = 3), and Reliable ($λ_1$ = 20, $λ_2$ =4) CG Execution Times are 1426.88 seconds and 1412.83 respectively. While this metric for Nominal is 750.48 seconds. We have added a statement regarding this on lines 337-339 in the revision. In terms of overhead, this execution time is less than 30 minutes, and since this planning is undertaken 4–5 months in advance, it is perfectly acceptable for real-world planning.
>
> - *Explanation, evaluation, and impact of Survival Models*: We perform our experiments on ~100k real-world flights in the US from the 2019 BTS dataset. This, to the best of our knowledge, is the first extensive evaluation for flight-crew pairing. The survival analysis models are trained with the standard 80:20 split, and we also report detailed metrics across the state-of-the-art survival analysis models (Tab 1). Since existing survival analysis metrics only care about the ranking or ordering (C-index, and Brier score), we also introduce P-index which measures how accurately our model can predict the exact time of landing – essential for real-world operations. Specifically, we perform extensive experimentation using three learning rates [0.0001,0.001,0.01], three batch sizes [16,32,64], and Monte Carlo simulations with 3 random seeds for each setting. We evaluate models using metrics such as Concordance, Within, Within_pct, MAE, and others.
>
> To evaluate the impact of the survival model, our reliable solutions reflect the value of our survival analysis-based data-driven approach as compared to the Nominal, which does not leverage the survival-based reliabilities. See Table 2, 3, 4, and Fig. 5 to see the value of survival analysis on the performance – leading to 61% improvements on the most challenging scenarios – an unprecedented advancement.
>
> While we provide all details about survival analysis and its experimental evaluation, we also provide pointers to further resources on survival analysis (line 193, [1], [2]) for an interested reader, since covering these would not be feasible.
> - *Theoretical Analysis*: We have not conducted a theoretical analysis of the framework. Nevertheless, we do consider challenging scenarios: regular and various levels of irregular operations to analyze the properties of SurvCG. Here, we find that the gains from SurvCG are especially significant when uncertainty is high (upper percentiles). See Tables 3 and 4. We may have slightly misunderstood what the reviewer means by “safety evaluation”, let us know if there are any additional questions.
> - *“Flights not only get delayed, but also arrive earlier than expected or get canceled”*: Here, we are assuming a push-back recovery strategy similar to the literature (Schaefer et al. 2005, Lu et al. 2015, Antunes et al. 2019). In this policy, delay is propagated by pushing the departure time of subsequent flights when needed. It is a commonly used in airlines in addition to other more extreme and more expensive policies like flight cancellations. It is reasonable to use a push-back policy at the planning stage, as the purpose is to come up with a crew pairing solution that can be recovered by push-back and not by other recovery policies. Incorporating these policies are interesting future research directions. The push-back policy assumption is clarified in the manuscript on page 6, lines 310-311.
>
> We sincerely appreciate the reviewer's valuable feedback, which has significantly strengthened our manuscript. Let us know if there are any additional questions, and we look forward to a further positive evaluation of our work.
>
> [1]  Dirk F Moore. Applied survival analysis using R, volume 473. Springer, 2016.
> [2]  David A Freedman. Survival analysis: A primer. The American Statistician, 62(2):110–119, 2008.

---

> > ### Author Response · Authors · 2024-12-03
> > **Official comment by authors**
> >
> > Dear Reviewer,
> >
> > Thank you for your valuable feedback on our paper. We've incorporated your suggestions into the revised rebuttal and would appreciate your thoughts on the changes and clarification. Please let us know if you need any further questions or have additional concerns. Hope this leads to a futher positive evaluation of our work.
> >
> > Thank you for your time and consideration.

---

### Meta-Review · Area_Chair_WZLF · 2024-12-20

**Metareview:**

This paper proposed a uncertainty-aware column generation method for crew pairing optimization. The key contribution is to use survival analysis for dynamic quantification of uncertainty based on historical data. Reviewers acknowledged the practical value this work, and the idea of using survival analysis for estimating missed crew connections in the context of airline crew scheduling. However, they also pointed out several key limitations, including: 1) insufficient descriptions about the solving procedure; 2) lacking experiments to support the advantage of survival analysis over other ML models such as regression; 3) lacking comparison to SOTA operations research baselines. These concerns still exist after rebuttal. While I think Reviewer yQi4's comment is a bit harsh (the topic is clearly related to ICLR, and theoretical analysis is welcome but not mandatory), however, I generally agree with their evaluation that this paper is below borderline. The proposed method is limited to a specific problem (crew pairing) with limited practical factors, and it is unclear whether it is generally applicable in column generation. While it could be difficult to implement some baselines, it is not a fully valid excuse to not compare with them since no code is available. Therefore, I recommend rejection.

**Additional Comments On Reviewer Discussion:**

Authors provided detailed point-to-point responses to all reviews. However, the responses are not sufficient to address all their concerns. In particular, reviewers requests more results (which is reasonable), but authors did not provide further results.

---

### Decision · Program_Chairs · 2025-01-22

Reject